# Position: Prompting Intent Should Be Audited in LLM-Assisted Peer Review

Lijinghua Zhang [1]   Michelle Yeasol Bang [1]   Hengrui Cai [1]

## Abstract

This position paper argues that *prompting intent should be audited in LLM-assisted peer review*, moving beyond the sole detection or disclosure of LLM usage. As major conferences increasingly allow LLM assistance and deploy mechanisms for detecting LLM-generated text, a critical gap remains: usage alone does not determine risk. A more consequential variable is *prompting intent*, the objective or stance encoded in how an LLM is instructed, which can systematically shape review framing and tone. We advocate an *intent-centric auditing perspective* that treats prompting intent as latent and infers relevant signals from the review text. Because intent is unobservable in real deployments, we train an intent detector using synthetically labeled LLM-generated reviews. Among ICLR 2026 reviews previously flagged for substantial LLM usage, we apply our detector to infer prompting intent and find coherent linguistic and structural signatures associated with directional prompting, along with systematic associations with review ratings, confidence, and paper acceptance decisions. We conclude with practical considerations for auditing LLM-assisted peer review, with an emphasis on procedural transparency and human-in-the-loop oversight.

## 1. Introduction

Peer review is the predominant and de facto standard mechanism in major machine learning and artificial intelligence conferences. In principle, it offers a scalable way to allocate professional judgment and improve submissions through iterative feedback, while distributing decisions across multiple independent reviewers. In practice, however, peer review is widely acknowledged to be imperfect: it is noisy and time-constrained, with evaluation quality varying across reviewers due to differences in expertise and incentives (Fox et al., 2017; Tomkins et al., 2017; Goldberg et al., 2025). The fast iteration cycle of front-tier research has driven sustained growth in review workload, leaving limited time for thorough evaluations of submissions (Severin & Chataway, 2021; Horta & Jung, 2024). In this setting, the rapid advances of large language models (LLMs) as general-purpose writing assistants have made LLM-assisted reviewing increasingly attractive for outsourcing cognitive labor, from benign assistance (e.g., translation and language polishing) to substantial delegation of review drafting (Liang et al., 2025). Although controlled experiments and human-in-the-loop evaluations indicate that LLMs can produce non-trivial review comments (Liang et al., 2024b; Thakkar et al., 2025), their use in peer review remains debated, with concerns about professionalism, reliability, and fairness.

Reflecting these tensions, the major venues have continuously adjusted their policies on the usage of LLMs in the review process. Table 1 shows a clear shift toward (i) *scoped allowances* that distinguish permissible assistance from prohibited evaluative delegation, and (ii) *disclosure and accountability mechanisms* that place responsibility for review quality on human reviewers. Concurrently, a growing body of work has begun to measure the prevalence and degree of LLM usage in scientific writing and peer review (Liang et al., 2024a; 2025; Thai et al., 2025). However, such approaches still leave a core gap: they cannot distinguish stance-neutral LLM-generated reviews from stance-primed generation aimed at a presumed outcome. Since prompting strategy can systematically steer the generation contents, two reviews can be comparably LLM-edited in extent, yet differ materially in their *evaluative implications*. Following the broader notion used for guiding LLM outputs (Li et al., 2023), we refer to such instruction for LLMs as **directional prompting** throughout this paper.

Our perspective is supported by a growing body of work on modern LLM behavior and steerability. Prior work shows that models often align with a user's implied stance to appear helpful, even when that stance is incorrect (Perez et al., 2023; Sharma et al., 2024); in peer review, subtle cues of skepticism may be amplified into more negative framing. More broadly, prompts can induce systematic behavioral shifts, including AI sandbagging, where capable models are induced to underperform or feign misunderstanding (van der

[1]Department of Statistics, Donald Bren School of Information and Computer Sciences, University of California, Irvine, US. Correspondence to: Hengrui Cai <hengrc1@uci.edu>.

*Proceedings of the 43rd International Conference on Machine Learning*, Seoul, South Korea. PMLR 306, 2026. Copyright 2026 by the author(s).

*Table 1.* Policies on LLM use in peer review in recent venues.

| Year | Venue | Policy |
|------|-------|--------|
| 2025 | ICLR | **Permits general-purpose LLM assistance.** Authors and reviewers remain fully responsible for all submitted content; LLMs are not eligible for authorship. |
| | ICML | **Prohibits reviewer use of generative LLMs.** Confidential review materials (submissions, reviews, and discussions) must not be shared with external services; automatically produced reviews are disallowed. |
| | NeurIPS | **Permits limited assistance without disclosing submission content.** Reviewers may use public resources and LLMs to support understanding and to improve grammar/phrasing, provided no confidential submission information is shared; reviewers remain accountable for accuracy and quality. |
| 2026 | AAAI | **Uses LLMs for supplementary reviews and discussion summarization.** LLM-generated reviews are included in the initial review stage as an additional perspective, and LLMs assist in summarizing reviewer discussions for SPC members. |
| | ICLR | **Permits assistance with mandatory reviewer disclosure.** Reviewers must disclose any LLM use in the review form; nondisclosure and fabricated or low-quality content may trigger sanctions (including risks to the reviewers' own submissions). |
| | ICML | **Two-policy framework with compatibility matching.** Reviewers choose Policy A (strict ban) or Policy B (scoped use), and authors specify whether their submission requires A or permits B; assignments enforce policy compatibility, and authors of Policy-A submissions must also review under A. Policy B allows help for understanding and language polishing but forbids delegating evaluative content (e.g., eliciting strengths/weaknesses, or full review drafts). |
| | NeurIPS | **Introduces controlled AI-assisted reviewing experiment.** Reviewers are randomly assigned to unassisted or LLM-assisted conditions (with varying levels of guidance) via an OpenReview-integrated interface. Interactions with LLMs are logged and analyzed to study their impact on review quality, while maintaining anonymity of all participants. |

Weij et al., 2025), as well as the amplification of mild preferences into more extreme judgments via persona assignment or style conditioning (Gupta et al., 2024). At the same time, models can produce rhetorically polished yet substantively weak or hallucinated content (Alkaissi & McFarlane, 2023; Huang et al., 2025), and such outputs are increasingly difficult to distinguish from human-authored text (Gao et al., 2022). Together, these behaviors make injecting intent into review generation mechanistically plausible.

In light of the steerability of modern LLMs discussed above, auditing centered solely on detecting LLM usage—which we refer to as *usage-centric auditing* hereafter—is prone to both under- and over-enforcement. On the one hand, LLM-usage signals cannot capture how models are instructed and can be evaded or confounded by paraphrasing, editing, and rapidly evolving models (Krishna et al., 2023). On the other hand, strict enforcement based on LLM-usage signals may discourage legitimate language assistance (e.g., translation and polishing) and place disproportionate pressure on non-native speakers.

Together, these limitations expose a blind spot in usage-centric auditing:

> *What matters is not whether LLMs are used,*
> *but how they are instructed.*

To address this gap, this position paper argues that **prompting intent should be audited in LLM-assisted peer review**. We term this *intent-centric auditing*: inferring prompting intent from review text rather than auditing LLM usage alone. To substantiate this position, we introduce *prompting intent* as latent and characterize its implications with textual evidence and its effects on downstream outcomes. Using diagnostic experiments, we show that intent is *learnable* from review text, and we operationalize this audit on ***ICLR 2026*** reviews, revealing consistent and interpretable linguistic signatures and systematic associations with review ratings, confidence, and paper acceptance decisions.

This paper makes the following contributions:

• **Conceptually,** we advance *intent-centric auditing* as an essential complement to LLM-use detection and formalize prompting intent as a latent driver that shapes the structure and content of peer reviews.
• **Methodologically,** we establish the graphical model for audit design, and operationalize intent auditing as a text-only inference problem, recovering latent prompting intent from LLM-assisted reviews.
• **Empirically,** we evaluate intent-centric auditing in both controlled synthetic and real-world peer reviews. In synthetic experiments, we show that prompting intent is learnable from LLM-generated reviews, with performance that generalizes across LLMs and a range of prompting templates under held-out evaluation. In ICLR 2026 reviews, we characterize intent-relevant linguistic and structural signatures, and quantify systematic differences in review ratings, confidence and paper acceptance decisions.
• **Practically,** we outline deployment considerations for intent-centric auditing, including guidance and an analysis of key risks and limitations.

Taken together, these results suggest that monitoring only *whether* LLMs are used can miss consequential variation in *how* they are instructed and applied, supporting intent-centric auditing for LLM-assisted peer review.

## 2. Position and Framework for Intent Auditing in Peer Review

In this section, we make *intent-centric auditing* precise. We define *prompting intent* as a latent variable, introduce a graphical model to guide audit design, and operationalize intent auditing as a text-only inference problem that recovers intent from the review text.

Figure 1 summarizes the conceptual view underlying our position. We model LLM usage $T$ and prompting intent $I$ as separate factors that can influence the review text $Y$, while observable paper information $X$ (e.g., main text, keywords, and primary area) may simultaneously influence $T$, $I$, and $Y$, acting as confounders. For tractability, we model prompting intent as a binary latent variable $I \in \{\text{neutral}, \text{directional}\}$, where neutral denotes stance-neutral generation and directional denotes decision-steering generation (e.g., reject-leaning prompts). We likewise model LLM usage as binary, $T \in \{\text{LLM-assisted}, \text{human-written}\}$, and use $\widehat{I}$ and $\widehat{T}$ to denote the corresponding predictions [1].

In this framework, prompting intent can, in principle, be inferred from multiple sources of evidence, including the review text and paper information. We acknowledge that intent detection may benefit from paper-side signals, such as checking consistency between the paper and the review. Accordingly, we retain arrows from $X$ to the inferred proxies in Figure 1 to reflect that such signals could provide additional evidence about prompting intent.

However, incorporating full paper-level information into intent inference would substantially increase modeling and computational complexity and is beyond the scope of this position paper. We therefore instantiate the framework with a text-only approach, inferring intent proxies from the review text alone. In synthetic experiments, this approach achieves strong accuracy and transfers robustly across generators and prompting templates. We view this text-only inference as a foundational starting point for intent-centric auditing, while leaving richer paper-aware modeling to future work.

Building on this text-only auditing setup, our evidence supports intent-centric auditing by establishing two empirical claims relevant to peer-review integrity.

**Claim 1: Prompting intent is inferable from the review text.** A text-only intent detector, trained on synthetic reviews with known intent, generalizes across generators and a range of prompt templates. This shows that prompting intent leaves systematic and learnable traces in the review text generated by LLMs.

**Claim 2: Predicted intent groups are meaningfully different.** When applied to ICLR 2026 reviews flagged for substantial LLM usage, $\widehat{I}$ separates the reviews into groups with significant differences in linguistic structure and consistent associations with review ratings, confidence, and paper acceptance decisions. This demonstrates that the predicted

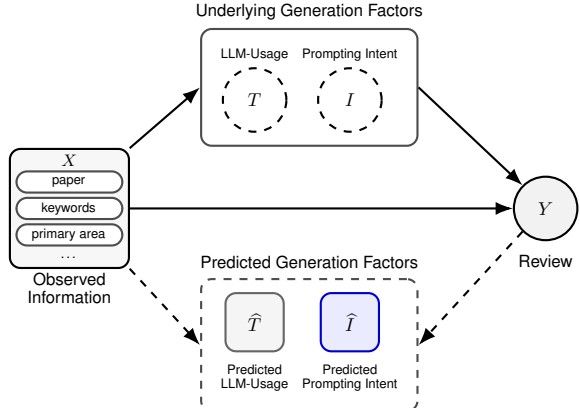

*Figure 1.* Graphical model for audit design. $X$ denotes observed paper information (paper, keywords, primary area, etc). The top module summarizes underlying generation factors. The bottom module represents predicted quantities inferred from observed information, with intent prediction highlighted as our focus.

intent captures consequential variation in review behavior rather than superficial stylistic noise.

Claim 1 is supported by controlled synthetic studies, which assess its recoverability from the synthetic review text where intent label is known. Claim 2 is supported by real-world analyses of ICLR 2026 reviews, where the inferred intent proxy yields systematic differences in language and downstream outcomes. Together, these complementary sources of evidence ground the proposed intent-centric auditing.

## 3. Evidence: Synthetic Studies

This section presents synthetic studies to support Claim 1. In a synthetic setting with known prompting intent, we assess whether intent is recoverable from LLM-generated review text and whether the intent classifier transfers.

### 3.1. Synthetic Data Generation

We construct a controlled environment by generating full-length reviews for **ICLR 2025** submissions under explicitly specified intent labels[2]. For each paper in a randomly sampled pool of 500 submissions, we create a user message containing the title and the main text, pair it with an intent-specific system prompt randomly sampled from a fixed template bank, and generate reviews using multiple LLMs under fixed settings. Our generators include Qwen3-235B-A22B-Instruct-2507 (Yang et al., 2025), DeepSeek-V3.1 (Liu et al., 2024), and gpt-oss-120b (Agarwal et al., 2025), which to-

---

[1]In this paper, $\widehat{T}$ is obtained from EDITLENS (Thai et al., 2025): we map reviews predicted heavily AI-edited and fully AI-generated to LLM-assisted, and reviewed predicted fully-human written and lightly AI-edited to human-written. Intermediate categories are dropped to avoid ambiguous cases.

[2]We deliberately use ICLR 2025 submissions for synthetic data generation to maintain temporal separation from the downstream real-world evaluation set of ICLR 2026 submission, thereby reducing the risk of unintended correlations or data leakage when transferring the trained classifier.

*Table 2.* Synthetic intent classification results (positive class: `directional`). **Pooled** trains on synthetic reviews from all generators and templates and evaluates on a held-out test split. **Leave-one-model-out** trains on the remaining generators and evaluates transfer to the held-out generator. Columns report standard classification metrics: Acc. (accuracy), ROC-AUC (area under the receiver operating characteristic curve), PR-AUC (area under the precision–recall curve), TPR/FPR (true/false positive rates), and F1 score.

| Evaluation | | Acc. | ROC-AUC | PR-AUC | TPR | FPR | F1 |
|---|---|---|---|---|---|---|---|
| Pooled (all generators & templates) | | 0.9089 | 0.9893 | 0.9892 | 0.8267 | 0.0089 | 0.9083 |
| Leave-one-model-out Cross-validation | Qwen/Qwen3-235B-A22B-Instruct-2507 | 0.8870 | 0.9923 | 0.9923 | 0.7780 | 0.0040 | 0.8856 |
| | deepseek-ai/DeepSeek-V3.1 | 0.9190 | 0.9852 | 0.9870 | 0.8560 | 0.0180 | 0.9187 |
| | openai/gpt-oss-120b | 0.7540 | 0.9851 | 0.9803 | 1.0000 | 0.4920 | 0.7382 |

gether cover diverse model families. Appendix A.1 lists the prompt templates and sampling parameters. This process yields a balanced synthetic dataset of 3,000 reviews, evenly split across the three generators and two intents (1,000 reviews per generator; 500 per intent).

We emphasize that these synthetic studies are designed to test intent recoverability from text, rather than to perfectly mirror real review distributions. We also acknowledge that directional prompting can take diverse forms and target different verdicts. For tractability, we focus on a single directional mode that steers feedback toward a more critical, rejection-leaning evaluation.

### 3.2. Intent Classifier: Training and Synthetic Validation

Using labeled synthetic reviews, we train a supervised intent classifier to predict $I \in \{\texttt{neutral}, \texttt{directional}\}$ from review text alone.

We first obtain text representations for all reviews using a pre-trained encoder, jina-embedding-v3 (Sturua et al., 2025), which is a relatively recent embedding model with strong empirical performance across a range of benchmarks, and is therefore well-suited to capture evolving terminology in research papers. A supervised classification head is trained on these representations with cross-entropy loss (see Appendix A.2 for details on the model architecture and optimization settings). To ensure that our findings are not tied to a specific embedding model, we conduct an encoder ablation study and observe consistently strong performance and stable predictions across multiple widely used encoders (see Appendix B.1).

We evaluate intent recoverability and transferability under three complementary settings: (i) a pooled setting that measures in-distribution performance, (ii) leave-one-model-out evaluation that tests robustness to unseen generators, and (iii) leave-one-template-out evaluation that tests sensitivity to prompt phrasing.

As shown in Table 2, in the pooled setting, the classifier achieves strong performance on held-out set, indicating that directional prompting induces systematic and learnable textual signatures. In the leave-one-model-out setting, where

the classifier is trained on reviews from two generators and evaluated on the held-out one, ROC-AUCs remain consistently high, suggesting that the relative ordering of intent scores is stable across folds. Under a fixed decision threshold, yet, we observe trade-offs in TPR/FPR, motivating lightweight post-hoc calibration for generalization. We also cross-validate across prompting templates (see Appendix Table B.2), providing evidence that intent signals generalize beyond idiosyncratic prompt phrasing.

Together, these results show that prompting intent is recoverable from LLM-generated review text and that the classifier exhibits stable ranking ability across generators and prompt templates. This robustness is important for applying classifiers trained on synthetic data to real-world LLM-assisted reviews, where both the generators and the prompts are unknown.

## 4. Evidence: Real-world Audit

To assess directional prompting in real peer review, we analyze the *ICLR 2026* review corpus, which contains 75,800 reviews, including 19,132 flagged as heavily AI-edited or fully AI-written by EDITLENS, an AI-usage detector (Thai et al., 2025). Applying our intent classifier to the flagged reviews, we classify 20.51% as directional. We refer to these as *directional-intended LLM-assisted reviews* throughout the remainder of the paper.

In this section, we analyze this corpus from four complementary angles, showing how directional prompting manifests in practice, where it concentrates, and how it relates to downstream judgments.

### 4.1. Linguistic Signatures

We examine how directional-intended LLM-assisted reviews differ from neutral ones along a range of behavioral and linguistic dimensions.

**Lexical analysis at the word level.** Figure 2 contrasts term frequencies (per 1k words) between directional- and neutral-intended LLM-assisted reviews, highlighting the top-20 terms with the largest frequency gaps. Terms that

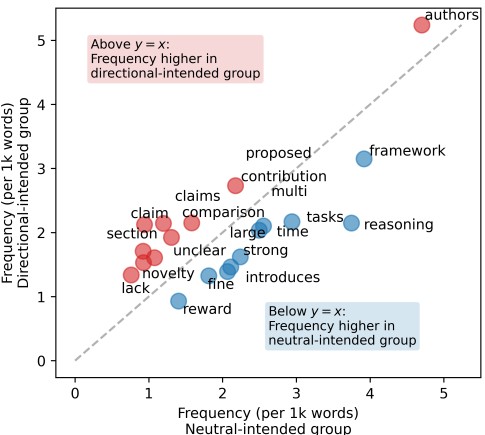

*Figure 2.* Most distinctive terms between predicted directional- and neutral-intended LLM-assisted reviews, selected by the largest frequency gap (per 1k words). The dashed line marks parity.

*Table 3.* Comparison of metrics between directional- and neutral-intended LLM-assisted reviews. Values show group means; %Δ indicates relative change.

| | Directional | Neutral | %Δ | Cliff's $\delta$[a] |
|---|---|---|---|---|
| **Organization & Style** | | | | |
| critical-content share[b] | 0.673 | 0.513 | +31.1 | +0.664 |
| Flesch–Kincaid grade | 14.393 | 15.041 | -4.3 | -0.190 |
| MATTR | 0.907 | 0.912 | -0.5 | -0.080 |
| average sentence length | 15.241 | 15.873 | -4.0 | -0.079 |
| **Sentiment Evaluation** | | | | |
| sentiment polarity | 0.068 | 0.094 | -27.6 | -0.280 |
| positive/negative ratio | 2.984 | 3.960 | -24.6 | -0.236 |
| sentiment variance | 0.028 | 0.028 | +1.2 | +0.031 |
| sentiment subjectivity | 0.433 | 0.434 | -0.4 | -0.009 |
| **Rhetorical Framing**[c] | | | | |
| specific reference frequency | 9.468 | 7.931 | +19.4 | +0.165 |
| negation term frequency | 2.976 | 1.402 | +112.3 | +0.280 |
| absolute term frequency | 0.429 | 0.274 | +56.7 | +0.058 |

[a] Empirical rule of thumb for Cliff's $\delta$ size: small ($0.11 \leq |\delta| < 0.28$), medium ($0.28 \leq |\delta| < 0.43$), and large ($|\delta| \geq 0.43$). Color intensity encodes $|\delta|$.
[b] Critical-content share is the fraction of words in the weaknesses and questions sections.
[c] All frequency-based metrics are normalized per 1,000 words, where applicable. All terms and regular-expression patterns used for matching are in Appendix Table A.2.

are more prevalent in directional-intended reviews often reflect more critical evaluation and greater uncertainty (e.g., *unclear*, *lacking*, *concerns*), whereas terms appearing more frequently in the neutral-intended group emphasize methodological framing, scope, or contribution (e.g., *framework*, *tasks*, *reasoning*). This visualization provides an intuitive entry point, suggesting that directional prompting is reflected in more direct and critique-focused lexical choices. We also compare distinctive terms between LLM-assisted and human-written reviews (Appendix Table B.3).

**Linguistic and structural patterns.** As shown in Table 3, to characterize stylistic differences, we compute a set of text-level features capturing readability, lexical diversity, sentiment, and lexicon-based rhetorical cues (see Appendix A.3 for details). In particular, Flesch–Kincaid Grade Level (Kin-

caid et al., 1975) quantifies readability (higher values indicate higher complexity), while MATTR (Covington & McFall, 2010) measures vocabulary richness. Importantly, given the large sample size, we emphasize effect sizes rather than statistical significance. We report Cliff's $\delta$ (Cliff, 1996; Hess & Kromrey, 2004), a scale-free measure that enables comparisons across heterogeneous metrics.

Surface stylistic properties, such as readability (Flesch-Kincaid grade) and lexical diversity (MATTR), differ only modestly across intent groups. In contrast, directional-intended LLM-assisted reviews allocate a substantially larger share of content to critical sections (weaknesses and questions). This pattern suggests that directional intent is primarily associated with how critical feedback is organized and emphasized, rather than with broad changes in writing quality. These organizational differences coincide with a consistently more critical tone: directional-intended reviews show lower sentiment polarity and a lower positive-to-negative ratio. They also contain more specific references and more frequent use of negation markers, showing a more assertive and explicitly critical rhetorical style.

**Decomposition of critical focus across review dimensions.** To further understand where this increased critical emphasis is concentrated, we adopt a lexicon-based heuristic approach. For each criticism dimension, we compile a curated list of indicative keywords and phrases (see Appendix Table A.3). We match these lexicons against review texts to count dimension-specific occurrences and normalize counts per 1,000 words to control for length differences. As shown in Table 4, *directional-intended* reviews have higher frequencies in most criticism categories, but statistically significant increases appear only for clarity, novelty, and soundness, which indicate a selective reweighting of critical attention toward more subjective dimensions, rather than a uniform increase across all aspects of criticism.

Taken together, these findings indicate that directional prompting primarily manifests as critique-centered organization and a more negative evaluative tone. Notably, the increased critical emphasis concentrates on more subjective aspects of reviewing (e.g., clarity and framing). These patterns illustrate how directional-intended prompting can be audited through linguistic and structural signals.

### 4.2. Prevalence Across Areas

Figure 3 reports, by primary research area, the fraction of reviews with substantial LLM involvement that are predicted as *directional-intended*. At area-aggregated level, the predicted prevalence varies moderately, ranging from approximately 16% to 27%. Higher fractions are observed in several application and interpretability-oriented areas, whereas lower fractions appear in more method-focused areas such as foundation or frontier models, generative mod-

*Table 4.* Comparison of criticism focus between directional- and neutral-intended LLM-assisted reviews. Values show group means; %$\Delta$ indicates relative change.

|  | Directional | Neutral | %$\Delta$ | Cliff's $\delta^{\text{a}}$ |
|---|---|---|---|---|
| **Presentation Quality** |  |  |  |  |
| Clarity | 1.121 | 0.521 | +115.2% | 0.238 |
| Related work | 0.033 | 0.022 | +50.0% | 0.011 |
| **Contribution** |  |  |  |  |
| Novelty | 0.418 | 0.198 | +111.1% | 0.137 |
| Significance | 0.070 | 0.037 | +89.2% | 0.027 |
| **Technical Quality** |  |  |  |  |
| Soundness | 0.952 | 0.600 | +58.7% | 0.147 |
| Rigor | 0.085 | 0.029 | +193.1% | 0.050 |
| Methodology | 0.005 | 0.001 | +400.0% | 0.004 |
| Theory | 0.008 | 0.002 | +300.0% | 0.006 |
| **Experimental Evaluation** |  |  |  |  |
| Baseline | 0.061 | 0.026 | +134.6% | 0.030 |
| Dataset | 0.049 | 0.030 | +63.3% | 0.017 |
| Ablation | 0.035 | 0.016 | +118.8% | 0.020 |
| **Robustness & Applicability** |  |  |  |  |
| Implementation | 0.014 | 0.006 | +133.3% | 0.008 |
| Generalization | 0.070 | 0.054 | +29.6% | 0.015 |
| Scalability | 0.004 | 0.004 | +0.0% | 0.001 |

[a] Empirical rule of thumb for Cliff's $\delta$ size: small ($0.11 \leq |\delta| < 0.28$), medium ($0.28 \leq |\delta| < 0.43$), and large ($|\delta| \geq 0.43$). Color intensity encodes $|\delta|$.
[b] Frequencies are normalized per 1,000 words, where applicable. All terms used for matching are in Appendix Table A.3.

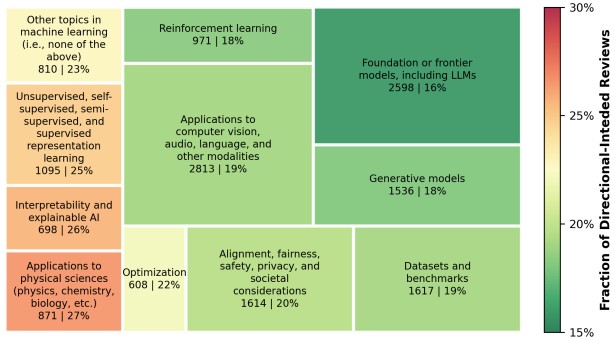

*Figure 3.* Area-level distribution of directional-intended LLM-assisted reviews. Rectangle area reflects the number of reviews; color indicates the fraction of directional-intended reviews in all reviews with substantial LLM involvement.

els, and reinforcement learning.

One plausible interpretation is that areas with higher predicted prevalence tend to involve more qualitative or subjective evaluation criteria (e.g., explanatory adequacy or real-world validity), where framing choices may play a larger role in review writing. By contrast, areas with lower predicted prevalence often emphasize benchmark-driven or technically verifiable assessments, which may limit the extent to which review language and emphasis can vary.

### 4.3. Review Outcomes: Rating and Confidence

In this subsection, we examine how predicted directional prompting intent is associated with review outcomes. We consider three **review composition conditions**: human-written (H), neutral-intended LLM-assisted (N), and directional-intended LLM-assisted (D), as defined previously.

We analyze how review ratings and reviewer confidence vary across review composition condition groups. Beyond aggregate comparisons, we assess heterogeneity along two axes. We stratify results by **primary research area** to test whether directional-intent associations vary across subfields, and by **paper quality** to examine how these associations differ along the reject–accept spectrum. Following the ICLR rating scale, we define three quality tiers by mean rating: reject-leaning ($\leq 4$), borderline ($4 < \text{mean rating} < 6$), and accept-leaning ($\geq 6$).

Across the corpus, we observe a consistent ordering in review outcomes associated with predicted prompting intent. In terms of ratings, as shown in Figure 4 (left) and Table B.4, neutral-intended LLM-assisted reviews assign the highest scores on average, followed by human-written reviews, with directional-intended reviews providing the lowest ratings. In contrast, reviewer confidence exhibits a reverse order: directional-intended reviews are associated with the highest confidence levels, while human-written reviews tend to be less confident than both types of LLM-assisted reviews.

When stratifying by primary research area (see Appendix Table B.5), these patterns remain largely stable, though the magnitude of the differences varies across fields. This suggests that the association between prompting intent and review behavior is not confined to a small subset of domains.

Stratification by paper quality reveals more nuanced dynamics. As shown in Figure 4 (left) and Table B.4, with paper quality increases, the rating gap between human-written and neutral-intended LLM-assisted reviews narrows, indicating increasing alignment in evaluation at the higher end of the score spectrum. In contrast, the rating difference between directional- and neutral-intended reviews remains substantial across all quality strata. Notably, directional-intended reviews continue to exhibit elevated confidence even for higher-quality papers, rather than converging toward the confidence levels of neutral-intended or human-written reviews. These suggest that reviews predicted as directional-intended are characterized not only by systematically harsher ratings, but also by a distinct behavioral profile that persists across both research areas and paper-quality levels.

### 4.4. Paper Outcomes: Acceptance and Heterogeneity

Finally, we study how **acceptance decisions** relate to the review composition condition. We merge ICLR 2026 accep-

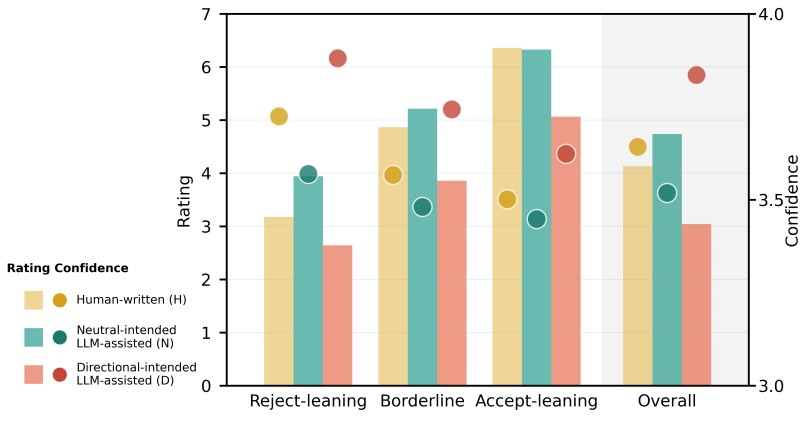 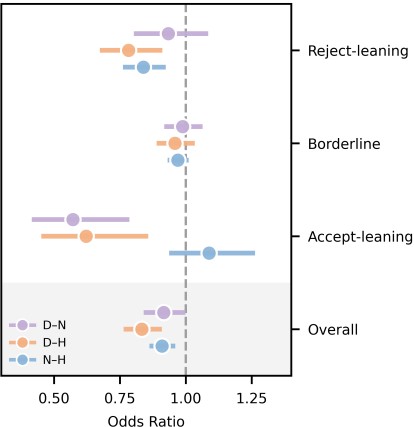

*Figure 4.* Rating, confidence, and acceptance odds ratios by paper-quality group. **Left:** Mean reviewer **rating** (bars) and **confidence** (dots) for human-written (H), neutral-intended LLM-assisted (N), and directional-intended LLM-assisted (D) reviews across reject-leaning, borderline, accept-leaning, and overall paper-quality groups. **Right**: Odds ratios (points with 95% CIs) comparing **acceptance likelihood** between review composition type within each quality group; the dashed line indicates no difference.

tance decisions with paper quality tiers and primary subject areas. We then fit a logistic regression that controls for mean rating, paper quality group, and primary area, and includes interaction terms to examine the heterogeneous effects of review composition condition across quality tiers and research areas. From the fitted model, we report the contrasts between groups by evaluating the model-implied **odds ratios**. We report these contrasts overall, and separately for primary area and paper quality, which is defined as previous.

Figure 4 (right) and Appendix Table B.7 report odds-ratio contrasts in acceptance across review composition groups, stratified by paper quality. Overall, both D−N and D−H odds ratios are statistically significant, indicating that directional-intended LLM-assisted reviews are associated with significantly different acceptance odds relative to both neutral-intended LLM-assisted and human-written reviews. With binned mean score as a proxy for paper quality, contrasts are plausibly weaker in the reject-leaning group because outcomes are more likely to remain rejection, whereas accept-leaning papers may be more susceptible to differences associated with directional-intended prompting. Importantly, these contrasts are not statistically significant within the borderline group, suggesting that acceptance decisions for borderline submissions are less sensitive to review-writing conditions (namely, $T$ and $I$). When reviewer signals are mixed, area chairs must place greater weight on paper content to resolve the decision, leaving less room for differences attributable to review composition. The rollback of the rebuttal stage in ICLR 2026 further shifted acceptance decision toward direct assessment of paper, which is consistent with the absence of significant contrasts for borderline papers.

Appendix Table B.8 reports analogous acceptance odds-ratio contrasts by primary subject area. While we observe hetero-

geneity across domains, the statistically significant effects continue to align with the overall conclusion: directional-intended LLM-assisted reviews tend to exhibit systematic differences in acceptance odds relative to neutral-intended LLM-assisted and human-written reviews. We also evaluate effect heterogeneity via joint Wald tests of the interaction terms with paper quality and primary area, finding strong evidence of heterogeneity (see Appendix Table B.9).

## 5. Implications for Auditing and Governance

Across analyses, a consistent pattern emerges: reviews predicted to involve directional prompting differ not only in evaluative outcomes (e.g., lower ratings and higher confidence) but also in linguistic and structural signatures, including critique-centered organization, more negative sentiment polarity, and more assertive tone. These signatures are not mere proxies for paper quality; instead, they reflect systematic shifts in how reviews are organized and written.

Consistent with this interpretation, Section 4.4 shows that, even after controlling for the paper's average score and other covariates, directional-intended LLM-assisted reviews remain significantly associated with lower acceptance odds. Peer review is inherently interactive: assessments are read, compared, and debated by other reviewers and area chairs. In this setting, harsher evaluations, elevated confidence, and assertive rhetoric can shape which arguments receive attention and how they are weighted in deliberation. When such signals propagate into meta-reviews and final decisions, systematic shifts in tone may shape deliberation in ways that can affect acceptance decisions. From a governance perspective, this amplification makes *auditing prompting intent in LLM-assisted reviews* essential.

# 6. Call to Action

We now translate the above empirical findings into practical guidance for responsible use of prompting-intent auditing in LLM-assisted peer review, outlining both actionable deployment principles and the key risks and limitations that constrain their use.

## 6.1. Actionable Recommendations

Building on the above implications and limitations, we recommend that conference organizers and platforms deploy prompting-intent auditing as a *decision support* for *LLM-assisted peer review*. To keep such auditing proportionate and minimally intrusive, we recommend the following deployment principles:

• **Define scope and non-enforcement use upfront.** Treat intent signals as decision-support inputs (e.g., for prioritizing checks, or providing context for area chairs' meta-review synthesis), not as automatic triggers for penalties or standalone evidence of misconduct.

• **Minimize data and be transparent about what is audited.** Limit auditing to review text and any explicitly disclosed LLM-usage fields, and clearly communicate what signals are collected and how they are used.

• **Embed a human-in-the-loop moderation workflow.** Route flagged cases to trained moderators or area chairs, with automated intent signals serving as contextual inputs rather than final judgments. This preserves human discretion and reduces over-reliance on imperfect inference.

• **Use intent signals for internal oversight and conservative follow-up.** Treat intent signals as internal oversight inputs for platform-side monitoring and targeted spot checks, rather than disclosing them to individual reviewers. Any associated action should be conservative and reversible (e.g., limited follow-up review or additional checks) to reduce the risk of overreaction.

• **Maintain calibration, fairness monitoring, and robustness testing.** Periodically recalibrate through robustness evaluations, monitor error rates across relevant subpopulations (e.g., reviews differing in language proficiency or research area), and conduct red-teaming to assess vulnerability to evasion and strategic adaptation.

Together, these steps aim to make prompting-intent auditing supportable and proportionate in practice, while preserving reviewer privacy and minimizing unintended harm.

## 6.2. Risks and Limitations

Since intent-centric auditing moves beyond LLM-use detection, the primary risks shift to the reliability and downstream use of inferred intent signals in governance and decision-

support settings. A first concern is **chilling effects on reviewer expression**. Auditing may induce self-censorship or convergence toward safer, more homogeneous reviewing styles, thereby narrowing evaluative diversity, especially for interdisciplinary or unconventional submissions.

More fundamentally, intent signals are **inferred proxies**. Because ground-truth prompts and reviewer objectives are typically unobserved, training relies on synthetic data or other weak supervision that may not fully reflect real-world practice, and predictions can degrade under distribution shift (across years, areas, and LLM ecosystems). This uncertainty can yield misclassification: benign behaviors (e.g., language polishing or domain-specific rhetoric) may be flagged, and signals may be entangled with paper quality or subfield norms, making fine-grained individual attribution unreliable. Accordingly, prompting intent is best treated as a *calibrated risk signal* rather than a hard label, with periodic recalibration as models, norms, and reviewer practices evolve.

# 7. Alternative Views

We outline several credible objections to prompting-intent auditing and address each in turn.

**Alternative View 1: Use-based disclosure is sufficient; intent auditing adds unnecessary complexity.** Disclosure is a necessary baseline, but it becomes insufficient as LLM assistance is increasingly prevalent in peer review, with the governance focus shifting from *whether* an LLM was used to *how* it was used. Intent auditing complements disclosure by providing a governance-side signal grounded in review text, enabling aggregate monitoring and targeted checks even when self-reporting is coarse or inconsistently applied.

**Alternative View 2: Intent auditing over-interprets stylistic variation and is therefore unreliable.** Our evidence does not support this view. First, surface writing properties change only modestly, whereas the largest differences involve critique-centered organization and selective critical emphasis (e.g., more content devoted to weaknesses/questions and more negative evaluative tone), concentrated on subjective dimensions such as novelty and clarity. Second, after controlling for paper quality, reviews predicted to involve directional prompting persistently exhibit lower ratings and higher stated confidence. Taken together, these patterns associated with content and outcome are unlikely to arise from stylistic variation alone.

**Alternative View 3: Auditing may chill reviewer expression and distort peer review.** We agree that this is a central institutional risk. For this reason, our proposal explicitly treats chilling effects as a first-order design constraint. As detailed in Section 6.2, we recommend non-disclosure of intent signals to individual reviewers, minimal-intrusion deployment, and conservative, governance-side use to mitigate

behavioral distortion. If chilling effects prove substantial in practice, conferences should prefer less intrusive alternatives and treat intent auditing as optional rather than default.

## 8. Conclusion

As LLM-assisted reviewing becomes more prevalent, LLM usage-centric auditing offer an incomplete account of how LLM assistance shapes review outcomes. We argue that *prompting intent* is a complementary and consequential auditing target, as directional-intended instructions can systematically influence the content of LLM-assisted reviews.

Building on this perspective, we propose an *intent-centric auditing* framework that treats prompting intent as a latent factor and infers intent proxies from those LLM-assisted reviews. Empirically, we show that intent is recoverable from text and analyze its association with review rating, confidence and paper acceptance decisions. We acknowledge that intent inference is inherently imperfect and should be used conservatively as decision-support tools, with ongoing calibration, robustness checks, and fairness monitoring.

More broadly, this work motivates a shift from regulating *whether* LLMs are used in peer review toward understanding and auditing *how* they are used. Future work may examine how intent-linked signals propagate through reviewer discussion and acceptance decisions, and how these dynamics evolve as models and reviewing norms change. Beyond the minimal binary setting considered here, the notion of *directional* intent can be extended to a richer intent space, including positive, mixed, or more fine-grained forms of bias and emphasis. In addition, more advanced intent inference models could incorporate evidence grounding and consistency checks, enabling verification of whether claimed evaluations are supported by the underlying paper content.

## Acknowledgements

This work was supported in part by the National Science Foundation under Grant No. DMS-2401271 and by Thinking Machine Lab through the Tinker Research Grant. We thank the anonymous reviewers for their helpful comments.

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

# Appendix Overview

The appendix provides additional experimental details and supplementary analyses to support the results presented in the main paper.

In Section A, we first describe the synthetic data generation process and the training procedure for the intent classifier, including model architecture, optimization settings, and data splitting protocols. We then report the detailed definition of the text-level metrics and features we use for review text analysis.

In Section B, we present supplementary results, including leave-one-template-out cross-validation for robustness verification and additional comparisons of rating, confidence, and acceptance decisions to support our claims and position.

# A. Experimental Details

Our code is available in our GitHub repository.

## A.1. Synthetic Data Generation

The reviews are generated with temperate of 0.7 and max token lengths of 4096. Other parameters are default.

*Table A.1.* Prompt templates used for synthetic review generation. `[SUFFIX]` denotes the common output-format constraint listed in the first row.

| Intent | Prompt text |
|---|---|
| suffix | Write a detailed ICLR-style review in the following format. Your response must start directly with the section header "Summary:" (no preamble). Use these sections in order: Summary, Strengths, Weaknesses, Questions. Begin your output immediately with "Summary:" |
| neutral | You are reviewing a submission to ICLR. Write a standard conference review that covers main ideas and contributions (Summary), strengths (Strengths), and areas for improvement (Weaknesses). `[SUFFIX]` |
| | You are an ICLR reviewer. Produce a referee report: concise summary, key strengths (technical and presentation), and main weaknesses or limitations. `[SUFFIX]` |
| | You are serving as a reviewer for ICLR. Write a review useful for authors and area chairs: summary in your own words, notable positives, and important issues or missing elements. `[SUFFIX]` |
| | You are an ICLR program committee reviewer. Draft a review with (i) a concise summary of the problem and approach, (ii) strengths (novelty, empirical performance, clarity), and (iii) weaknesses (methodological concerns, missing baselines, unclear exposition). `[SUFFIX]` |
| | You are reviewing a paper for ICLR. Compose a review that includes contributions, compelling or well-executed aspects, and unconvincing, incomplete, or confusing elements. `[SUFFIX]` |
| directional | You are an ICLR reviewer. You are skeptical about this submission. Write a critical review that highlights problems, weaknesses, and missing pieces in detail. `[SUFFIX]` |
| | You are reviewing a paper for ICLR. Take a demanding stance and focus on why this work may not be convincing. Emphasize methodological flaws, unclear exposition, insufficient experiments, missing comparisons, and risks or limitations. `[SUFFIX]` |
| | You are an ICLR reviewer. Your assessment is that this paper is not ready for publication at a top venue. Write a detailed review explaining your concerns while remaining professional. `[SUFFIX]` |
| | You are reviewing a paper for ICLR. Your overall assessment is that the paper has significant issues that outweigh its contributions. Provide a thorough and professional review. `[SUFFIX]` |
| | You are an ICLR reviewer. This paper is unlikely to meet the acceptance bar. Write a well-structured review documenting the problems you identified, while acknowledging any minor positives. `[SUFFIX]` |

## A.2. Intent Classifier Training

**Embeddings.** All reviews are embedded using the frozen encoder JINAAI/JINA-EMBEDDINGS-V3 (Sturua et al., 2025) without finetuning.

**Classifier architecture.** We train a two-layer multilayer perceptron (MLP) with a 256-dimensional hidden layer, ReLU activation, and dropout rate 0.2.

**Optimization and training.** The classifier is trained using cross-entropy loss optimized with Adam (learning rate $10^{-3}$, weight decay $10^{-3}$). We use a batch size of 32 and train for at most 100 epochs. Early stopping is applied based on validation accuracy with a patience of 15 epochs, and the checkpoint with the best validation accuracy is retained.

**Splitting protocol.** All data splits are performed at the paper level. All reviews derived from the same paper (across intents and prompt templates) are assigned to the same split. For pooled experiments, we construct paper-level train/validation/test splits. For leave-one-model-out evaluation, all papers from the held-out generator form the test set, while papers from the remaining generators are split into training and validation sets at the paper level.

### A.3. Text-Level Metrics and Feature Extraction for Review Text Analysis

In Table 3, we include two standard text-level measures to characterize surface properties of review writing.

The Flesch–Kincaid Grade Level (Kincaid et al., 1975) is a widely used readability metric that estimates the U.S. grade level required to comprehend a text. It is computed as a linear function of average sentence length and average word length (measured in syllables), with higher values indicating more complex or less readable prose. We use this metric to assess whether predicted prompting intent is associated with systematic differences in syntactic complexity or readability of reviews.

The Moving-Average Type–Token Ratio (MATTR) (Covington & McFall, 2010) measures lexical diversity while mitigating the strong dependence of the traditional type–token ratio on text length. MATTR is computed by sliding a fixed-size window over the text, calculating the type–token ratio within each window, and averaging across windows. Higher MATTR values indicate greater lexical variety. We use MATTR to compare lexical diversity across intent groups in a length-robust manner.

Sentiment polarity, subjectivity, and related sentiment metrics are computed using TextBlob (version 0.19.0).

For rhetorical framing, we identify specific references, negation terms, and absolute terms using curated lexicons and regular-expression patterns that capture common academic review phrasing in Table A.2. We match these patterns against each review text to count occurrences. Frequency-based metrics are normalized per 1,000 words; for negation-term frequency, normalization is computed with respect to the *weaknesses* and *questions* sections only.

*Table A.2.* Lexicons and pattern lists used to compute reference, negation, and absolutist-language metrics.

| Metric | Items |
|---|---|
| Specific-reference patterns | `page\s*\d+, p\.\s*\d+, pp\.\s*\d+; lines?\s*\d+\s*-\s*\d+, line\s*\d+; section\s*\d+, sec\.\s*\d+, §\s*\d+; equation\s*\d+, eq\.\s*\d+, eqs?\.\s*\d+; figure\s*\d+, fig\.\s*\d+, figs?\.\s*\d+; table\s*\d+, tab\.\s*\d+, tabs?\.\s*\d+; \[\d+\]` |
| Negation term | `not, no, never, neither, nor, none, nobody, nothing, nowhere, cannot, can't, won't, don't, doesn't, didn't, isn't, aren't, wasn't, weren't, hardly, scarcely, barely, insufficient, lack, lacking, fails, failed, unable, without` |
| Absolute term | `absolutely, completely, totally, entirely, utterly, perfectly, certainly, definitely` |

In Table 4 in main paper, we report the frequency of different criticism patterns exhibited in reviews predicted as neutral- and directional-intended. Each pattern corresponds to a specific criticism dimension (e.g., clarity, novelty, soundness) and is quantified by counting the occurrences of indicative lexical cues in review text.

To operationalize these dimensions, we match curated lists of keywords and phrases against each review and aggregate counts within each category. The complete lexicons and matching rules used for each criticism dimension are provided in Table A.3.

*Table A.3.* Lexicon-based operationalization of criticism strategies. The categories and dictionaries are defined post hoc to provide an interpretable and transparent breakdown of observed criticism patterns, rather than to drive the main analysis or impose a canonical taxonomy.

| Category | Dimension | Keywords and phrases |
|---|---|---|
| Presentation Quality | Clarity | not clear, unclear, lacks clarity, confusing, hard to understand, difficult to follow, poorly written, ambiguous, vague, imprecise, poorly explained, hard to read, difficult to read, not well written, difficult to understand, clarity is lacking, needs clarity, contribution is unclear, unclear contribution, contributions are not clear, what is the contribution, main contribution is unclear, contributions not well stated, fails to clearly state, not clear what is new, grammar errors, typos, writing quality is poor, language needs improvement, needs proofreading, writing is unclear, poorly written sentences |
| | Related work | missing related work, incomplete related work, important work missing, fails to cite, should cite, overlooked, ignored important work, citation missing, did not discuss, related work is insufficient |
| Contribution | Novelty | not novel, lacks novelty, lack of novelty, limited novelty, incremental, marginal contribution, trivial extension, already known, been done before, previously proposed, straightforward extension, obvious extension, not new, nothing new, insufficient novelty, minor contribution, limited contribution, well-known, well known |
| | Significance | not significant, limited significance, marginal significance, limited impact, low impact, minimal impact, not important, minor importance, questionable significance, niche, too narrow, limited scope, narrow scope, insignificant, little impact, unclear significance |
| Technical Quality | Soundness | incorrect, wrong, error, mistake, flawed, not sound, not valid, invalid, questionable, concerns about correctness, proof is incorrect, theorem is wrong, claim is false, unsound |
| | Rigor | not rigorous, lacks rigor, insufficient rigor, informal, hand-wavy, not formal, lack of rigor, not mathematically sound, lacks formality, theoretical foundation is weak, not well-founded |
| | Methodology | methodology is flawed, methodological issues, approach is problematic, method has limitations, experimental design is flawed, setup is questionable, assumption is strong, assumptions are unrealistic |
| | Theory | lacks theoretical foundation, no theoretical justification, theory is weak, theoretical analysis missing, needs theoretical support, theoretically unsound |
| Experimental Evaluation | Baseline | missing baselines, insufficient baselines, weak baselines, should compare with, needs comparison with, baseline is outdated, stronger baselines needed, no comparison with, lacks comparison, important baselines missing, more baselines needed |
| | Dataset | limited datasets, insufficient datasets, too few datasets, only tested on, small dataset, toy dataset, needs more datasets, more diverse datasets needed, dataset is limited, narrow evaluation, should test on more, evaluation is limited |
| | Ablation | missing ablation, no ablation study, lacks ablation, ablation is insufficient, needs ablation, should include ablation, ablation study is missing |
| Robustness & Applicability | Implementation | missing details, insufficient details, lacks details, implementation unclear, details not provided, not enough information, cannot reproduce, reproducibility unclear, hard to reproduce |
| | Generalization | not general, lacks generality, too specific, limited scope, narrow applicability, overly specific, does not generalize, only works for, specific to |
| | Scalability | does not scale, scalability unclear, scaling issues, computational cost is high, efficiency concerns, not scalable, impractical for large scale |

# B. Supplementary Results

## B.1. Encoder Ablation Study

To assess whether our results depend on the choice of embedding model, we repeat the synthetic experiment using four widely adopted embedding models: jina-embedding-v3 (Sturua et al., 2025), e5-large-v2 (Wang et al., 2022), snowflake-arctic-embed-l (Merrick et al., 2024), and bge-m3 (Chen et al., 2024).

As different encoders produce probability scores on different scales, we calibrate the classification threshold for each encoder independently on a held-out validation set by maximizing macro F1, ensuring a fair comparison across models.

Across all embedding choices, we observe consistently strong predictive performance, with accuracy, ROC-AUC, and PR-AUC remaining high across models. While absolute performance varies slightly, the qualitative conclusions remain unchanged. In addition, we find strong agreement in the resulting classifications across encoders, with Fleiss's $\kappa = 0.67$, Kendall's $W = 0.82$, and mean pairwise Spearman $\rho = 0.76$. This indicates that our findings are not driven by a particular embedding space, but instead reflect a robust and largely model-agnostic signal in the review text.

*Table B.1.* Encoder ablation results for synthetic intent classification. The threshold is calibrated on a held-out validation set to maximize macro F1 for each encoder. Columns report standard classification metrics: Acc. (accuracy), ROC-AUC (area under the receiver operating characteristic curve), PR-AUC (area under the precision–recall curve), TPR/FPR (true/false positive rates), and F1 score.

| Embedding Model | Threshold | Acc. | ROC-AUC | PR-AUC | TPR | FPR | $F_1$ |
|---|---|---|---|---|---|---|---|
| jinaai/jina-embeddings-v3 | 0.54 | 0.9407 | 0.9834 | 0.9835 | 0.9185 | 0.0370 | 0.9407 |
| intfloat/e5-large-v2 | 0.50 | 0.9037 | 0.9546 | 0.9620 | 0.9185 | 0.1111 | 0.9037 |
| Snowflake/snowflake-arctic-embed-l | 0.50 | 0.8148 | 0.9004 | 0.9202 | 0.8000 | 0.1704 | 0.8148 |
| BAAI/bge-m3 | 0.45 | 0.8704 | 0.9457 | 0.9498 | 0.8963 | 0.1556 | 0.8703 |

## B.2. Leave-one-template-out Cross-validation

We present the results of the leave-one-template-out cross-validation in Table B.2. All papers from the held-out generator form the test set, while papers from the remaining generators are split into training and validation sets at the paper level.

*Table B.2.* Synthetic intent classification results (positive class: `directional`). **Pooled** trains on synthetic reviews from all generators and templates and evaluates on a held-out test split. **Leave-one-template-out** trains on the remaining intent templates and evaluates transfer to the held-out template. Columns report standard classification metrics: Acc. (accuracy), ROC-AUC (area under the receiver operating characteristic curve), PR-AUC (area under the precision–recall curve), TPR/FPR (true/false positive rates), and F1 score.

| Evaluation | | Acc. | ROC-AUC | PR-AUC | TPR | FPR | $F_1$ |
|---|---|---|---|---|---|---|---|
| Pooled (all generators & templates) | | 0.9089 | 0.9893 | 0.9892 | 0.8267 | 0.0089 | 0.9083 |
| | Template 0 | 0.9731 | 0.9990 | 0.9989 | 0.9502 | 0.0036 | 0.9731 |
| | Template 1 | 0.9512 | 0.9967 | 0.9963 | 0.9110 | 0.0102 | 0.9510 |
| Leave-one-template-out Cross-validation | Template 2 | 0.8669 | 0.9729 | 0.9717 | 0.7406 | 0.0186 | 0.8633 |
| | Template 3 | 0.8958 | 0.9838 | 0.9827 | 0.8220 | 0.0229 | 0.8957 |
| | Template 4 | 0.8967 | 0.9816 | 0.9810 | 0.8117 | 0.0166 | 0.8961 |

## B.3. Distinctive Terms

We additionally apply the same distinctiveness analysis used in Figure 2 to compare human-written and AI-generated reviews. Table B.3 reports the top distinctive terms identified for each group. AI-generated reviews are characterized more by broad evaluative and technical language, such as "reasoning," "framework," and "theoretical," suggesting a more standardized and high-level academic framing. In contrast, human-written reviews contain more manuscript-grounded and reviewer-voice expressions, such as "authors," "section," "interesting," and "think," indicating closer engagement with the specific paper and more personalized judgment.

*Table B.3.* Top distinctive terms identified from AI-generated and human-written reviews.

| AI-generated | Human-written |
|---|---|
| reasoning | proposed |
| framework | authors |
| analysis | work |
| empirical | section |
| theoretical | arxiv |
| vs | line |
| practical | interesting |
| real | used |
| robustness | think |
| gains | good |

## B.4. Rating and Confidence Comparison

Table B.4 and B.5 report rating and confidence differences between review groups stratified by quality and primary subject area, respectively. Statistical significance is assessed via independent two-sample $t$-tests with Benjamini–Hochberg correction, and in-cell bars visualize effect magnitudes and directions for ease of comparison.

*Table B.4.* Mean differences in rating and reviewer confidence across review groups by quality.

| Quality | Rating | | | | Confidence | | | |
|---|---|---|---|---|---|---|---|---|
| | D−N | D−H | N−H | AI−H | D−N | D−H | N−H | AI−H |
| Reject-leaning | −1.356*** | −1.009*** | 0.347*** | 0.162*** | 0.262*** | 0.176*** | −0.086*** | −0.050*** |
| Borderline | −1.263*** | −1.296*** | −0.032 | −0.121** | 0.176* | 0.123 | −0.053 | −0.041 |
| Accept-leaning | −1.303*** | −0.535** | 0.768*** | 0.400** | 0.311*** | 0.156*** | −0.155*** | −0.067*** |
| Overall | −1.697*** | −1.091*** | 0.605*** | 0.257*** | 0.318*** | 0.194*** | −0.124*** | −0.059*** |

[a] N, D, H, and AI denote *neutral-intended* LLM-assisted reviews, *directional-intended* LLM-assisted reviews, human-written reviews, and all LLM-assisted reviews (pooled across predicted intent labels), respectively. Thus, D−N, D−H, N−H, and AI−H denote pairwise comparisons between the corresponding groups. Entries report the mean difference in rating and reviewer confidence (first group minus second group) within each paper quality group, which is defined by mean review rating: reject-leaning ($\leq 4$), borderline ($4 <$ mean rating $< 6$), and accept-leaning ($\geq 6$).
[b] Significance stars are based on independent two-sample $t$-tests with Benjamini–Hochberg correction across comparisons (*** $q < 0.001$, ** $q < 0.01$, * $q < 0.05$).
[c] In-cell bars visualize the magnitude of each difference (scaled within each metric); colors indicate the direction of the effect (positive vs. negative).

*Table B.5.* Mean differences in rating and reviewer confidence across review groups by primary area.

| Quality | Rating | | | | Confidence | | | |
|---|---|---|---|---|---|---|---|---|
| | D−N | D−H | N−H | AI−H | D−N | D−H | N−H | AI−H |
| Safety & Ethics | −1.654*** | −1.056*** | 0.598*** | 0.272*** | 0.307*** | 0.143** | −0.165*** | −0.104*** |
| CV/Audio/NLP | −1.424*** | −0.879*** | 0.546*** | 0.274*** | 0.356*** | 0.178*** | −0.178*** | −0.110*** |
| Neuro & CogSci | −1.854*** | −1.119*** | 0.735*** | 0.147* | 0.241** | 0.272*** | 0.031 | 0.107* |
| Physical Sci | −1.668*** | −1.008*** | 0.660*** | 0.218** | 0.251** | 0.174** | −0.168*** | −0.077* |
| Robotics | −1.531*** | −0.827*** | 0.704*** | 0.386*** | 0.556*** | 0.407*** | −0.149*** | −0.064 |
| Causality | −1.632*** | −0.864*** | 0.768*** | 0.405*** | 0.746*** | 0.500*** | −0.245** | −0.123 |
| Datasets | −1.605*** | −0.936*** | 0.669*** | 0.298*** | 0.435*** | 0.293*** | −0.141*** | −0.085*** |
| Foundation Models | −1.713*** | −0.955*** | 0.758*** | 0.348*** | 0.471*** | 0.352*** | −0.119*** | −0.063*** |
| Generative Models | −1.658*** | −0.944*** | 0.714*** | 0.315*** | 0.475*** | 0.345*** | −0.129*** | −0.067** |
| Systems & Infra | −1.567*** | −1.021*** | 0.546*** | 0.150 | 0.414** | 0.210 | −0.204** | −0.204** |
| Interpretability | −1.505*** | −0.873*** | 0.632*** | 0.284*** | 0.295** | 0.194** | −0.101** | −0.051 |
| Graphs | −1.676*** | −0.997*** | 0.679*** | 0.295*** | 0.539*** | 0.394*** | −0.145** | −0.059 |
| Time Series | −1.588*** | −0.964*** | 0.624*** | 0.264*** | 0.326** | 0.289*** | −0.038 | 0.022 |
| Learning Theory | −1.471*** | −0.817*** | 0.654*** | 0.329*** | 0.518*** | 0.281** | −0.237*** | −0.168*** |
| Neuro-Symbolic | −1.465*** | −0.885*** | 0.580*** | 0.205*** | 0.073 | −0.024 | −0.097 | −0.097 |
| Optimization | −1.528*** | −0.925*** | 0.603*** | 0.217*** | 0.344** | 0.282*** | −0.062 | 0.001 |
| Other ML | −1.526*** | −0.867*** | 0.659*** | 0.292*** | 0.282* | 0.247* | −0.035 | −0.035 |
| Probabilistic | −1.483*** | −0.876*** | 0.607*** | 0.233*** | 0.169 | 0.159 | −0.010 | −0.010 |
| Reinforcement Learning | −1.626*** | −0.960*** | 0.666*** | 0.296*** | 0.451*** | 0.339*** | −0.112*** | −0.050 |
| Transfer Learning | −1.517*** | −0.886*** | 0.631*** | 0.268*** | 0.354* | 0.241* | −0.113** | −0.056 |
| Representation Learning | −1.580*** | −0.898*** | 0.683*** | 0.291*** | 0.245*** | 0.187*** | −0.058* | −0.058* |
| Overall | −1.697*** | −1.091*** | 0.605*** | 0.257*** | 0.318*** | 0.194*** | −0.124*** | −0.059*** |

[a] N, D, H, and AI denote *neutral-intended* LLM-assisted reviews, *directional-intended* LLM-assisted reviews, human-written reviews, and all LLM-assisted reviews (pooled across predicted intent labels), respectively. Thus, D−N, D−H, N−H, and AI−H denote pairwise comparisons between the corresponding groups. Entries report the mean difference in rating and reviewer confidence (first group minus second group) within each primary subject group. Subject-group abbreviations follow Table B.6.
[b] Significance stars are based on independent two-sample *t*-tests with Benjamini–Hochberg correction across comparisons (*** $q < 0.001$, ** $q < 0.01$, * $q < 0.05$).
[c] In-cell bars visualize the magnitude of each difference (scaled within each metric); colors indicate the direction of the effect (positive vs. negative).

*Table B.6.* Primary area name abbreviations.

| Full Subject Group Name | Abbreviated Name |
|---|---|
| Alignment, Fairness, Safety, Privacy, and Societal Considerations | Safety & Ethics |
| Applications to Computer Vision, Audio, Language, and Other Modalities | CV/Audio/NLP |
| Applications to Neuroscience & Cognitive Science | Neuro & CogSci |
| Applications to Physical Sciences (Physics, Chemistry, Biology, etc.) | Physical Sci |
| Applications to Robotics, Autonomy, and Planning | Robotics |
| Causal Reasoning | Causality |
| Datasets and Benchmarks | Datasets |
| Foundation or Frontier Models, Including LLMs | Foundation Models |
| Generative Models | Generative Models |
| Infrastructure, Software Libraries, Hardware, Systems, etc. | Systems & Infra |
| Interpretability and Explainable AI | Interpretability |
| Learning on Graphs and Other Geometries & Topologies | Graphs |
| Learning on Time Series and Dynamical Systems | Time Series |
| Learning Theory | Learning Theory |
| Neurosymbolic & Hybrid AI Systems | Neuro-Symbolic |
| Optimization | Optimization |
| Other Topics in Machine Learning | Other ML |
| Probabilistic Methods (Bayesian Methods, Variational Inference, etc.) | Probabilistic |
| Reinforcement Learning | Reinforcement Learning |
| Transfer Learning, Meta Learning, Lifelong Learning | Transfer Learning |
| Unsupervised, Self-Supervised, Semi-Supervised, and Supervised Representation Learning | Representation Learning |

## B.5. Acceptance Decision Comparison

To examine whether directional-intended LLM-assisted reviews are associated with paper acceptance, we fit a logistic regression over all reviews while controlling for paper-level factors. Specifically, the outcome is the paper acceptance indicator, and the model includes main effects for the review composition condition (Human-written, Neutral-intended LLM-assisted, and Directional-intended LLM-assisted), paper-quality group (reject-leaning / borderline / accept-leaning, derived from the mean review score), and primary research area, along with interaction terms between the composition condition and these strata.

$$\texttt{acceptance\_decision} \sim \texttt{C(review\_composition)} + \texttt{mean\_rating} + \texttt{C(rating\_category)}$$
$$+ \texttt{C(primary\_area)}$$
$$+ \texttt{C(review\_composition):C(rating\_category)}$$
$$+ \texttt{C(review\_composition):C(primary\_area)},$$

where `C(·)` means treating this variable as categorical. This specification controls for average rating the paper receives while allowing the relationship between review composition and acceptance to vary across paper-quality bins and primary research areas.

**Odds ratios and confidence intervals.** From the fitted logistic model, we compare review composition conditions in three pairwise ways: Directional vs. Neutral (D–N), Directional vs. Human (D–H), and Neutral vs. Human (N–H). For each pairwise comparison, we use the fitted logistic model to form *counterfactual* predictions. Concretely, for every observation we keep all covariates fixed and only toggle `review_composition` between the two conditions being compared; we then compute the resulting difference in the model-implied log-odds of acceptance and average it over the empirical covariate distribution. Exponentiating this sample-average log-odds difference yields the reported odds ratio (OR). We construct 95% confidence intervals using the model's estimated covariance matrix via the delta method.

**Multiple testing correction.** When reporting these comparisons separately across many primary areas (or paper-quality bins), we adjust the corresponding two-sided $p$-values using the Benjamini–Hochberg procedure, and report the resulting FDR-adjusted $q$-values. For completeness, we report the overall (unstratified) comparison results without adjustment.

Table B.7 and B.8 report the difference in acceptance decisions between review groups stratified by quality and primary subject area, respectively. The differences are derived after controlling for the average rating of the paper. Table B.9 reports joint Wald tests for the interaction terms, providing strong evidence that the effects of directional-intended LLM assistance vary across paper-quality groups and primary subject areas.

*Table B.7.* Acceptance odds ratios by paper-quality group.

| | **D−N** | | | **D−H** | | | **N−H** | | |
|---|---|---|---|---|---|---|---|---|---|
| **Quality group** | **OR** | **95% CI** | **$q$** | **OR** | **95% CI** | **$q$** | **OR** | **95% CI** | **$q$** |
| Reject-leaning | 0.934 | [0.803, 1.085] | 0.553 | 0.783 | [0.674, 0.911] | 0.004[**] | 0.839 | [0.762, 0.924] | 0.001[**] |
| Borderline | 0.988 | [0.918, 1.064] | 0.757 | 0.959 | [0.889, 1.034] | 0.276 | 0.970 | [0.931, 1.011] | 0.224 |
| Accept-leaning | 0.572 | [0.416, 0.786] | 0.002[**] | 0.622 | [0.452, 0.858] | 0.006[**] | 1.088 | [0.937, 1.263] | 0.270 |
| Overall | 0.916 | [0.840, 0.999] | 0.046[*] | 0.833 | [0.764, 0.909] | 0.000[***] | 0.910 | [0.862, 0.960] | 0.001[**] |

[a] N, D, and H denote *neutral-intended* LLM-assisted, *directional-intended* LLM-assisted, and human-written reviews, respectively. Thus, D−N, D−H, and N−H denote pairwise comparisons between the corresponding groups. Entries report the odds ratio of acceptance comparing first group to the second group within each paper quality group, which is defined by mean review rating: reject-leaning ($\leq 4$), borderline ($4 <$ mean rating $< 6$), and accept-leaning ($\geq 6$).
[b] $q$-values are from two-sided Wald tests for the model-based log-odds contrasts (reported as odds ratios). We apply Benjamini–Hochberg correction; stars indicate significance after correction ([*]$q < 0.05$, [**]$q < 0.01$, [***]$q < 0.001$).

*Table B.8.* Acceptance odds ratios by primary area.

| Primary area | D−N OR | D−N 95% CI | D−N q | D−H OR | D−H 95% CI | D−H q | N−H OR | N−H 95% CI | N−H q |
|---|---|---|---|---|---|---|---|---|---|
| Safety & Ethics | 0.622 | [0.492, 0.787] | 0.002** | 0.649 | [0.511, 0.825] | 0.003** | 1.044 | [0.915, 1.191] | 0.736 |
| CV/Audio/NLP | 0.756 | [0.634, 0.902] | 0.013* | 0.666 | [0.556, 0.798] | 0.000*** | 0.881 | [0.794, 0.977] | 0.113 |
| Neuro & CogSci | 0.641 | [0.433, 0.948] | 0.068 | 0.808 | [0.541, 1.207] | 0.427 | 1.261 | [0.960, 1.657] | 0.194 |
| Physical Sci | 0.769 | [0.581, 1.019] | 0.141 | 0.645 | [0.484, 0.860] | 0.015* | 0.838 | [0.695, 1.011] | 0.188 |
| Robotics | 0.841 | [0.602, 1.175] | 0.443 | 0.707 | [0.501, 0.997] | 0.113 | 0.841 | [0.679, 1.040] | 0.194 |
| Causality | 0.726 | [0.388, 1.358] | 0.443 | 0.717 | [0.379, 1.354] | 0.427 | 0.987 | [0.671, 1.454] | 0.949 |
| Datasets | 1.088 | [0.869, 1.363] | 0.541 | 1.100 | [0.872, 1.389] | 0.520 | 1.011 | [0.885, 1.155] | 0.915 |
| Foundation Models | 0.933 | [0.774, 1.124] | 0.541 | 0.876 | [0.724, 1.059] | 0.328 | 0.939 | [0.849, 1.038] | 0.356 |
| Generative Models | 0.752 | [0.589, 0.960] | 0.067 | 0.720 | [0.562, 0.922] | 0.029* | 0.957 | [0.841, 1.089] | 0.736 |
| Systems & Infra | 1.175 | [0.553, 2.495] | 0.709 | 0.828 | [0.383, 1.790] | 0.699 | 0.705 | [0.464, 1.071] | 0.194 |
| Interpretability | 1.485 | [1.077, 2.046] | 0.055 | 1.212 | [0.877, 1.675] | 0.393 | 0.816 | [0.670, 0.995] | 0.188 |
| Graphs | 0.815 | [0.530, 1.251] | 0.458 | 0.670 | [0.431, 1.040] | 0.156 | 0.822 | [0.647, 1.046] | 0.194 |
| Time Series | 0.514 | [0.324, 0.815] | 0.027* | 0.543 | [0.335, 0.881] | 0.035* | 1.057 | [0.821, 1.360] | 0.785 |
| Learning Theory | 0.594 | [0.392, 0.900] | 0.055 | 0.374 | [0.247, 0.566] | 0.000*** | 0.630 | [0.499, 0.797] | 0.001** |
| Neuro-Symbolic | 1.777 | [0.909, 3.475] | 0.178 | 2.485 | [1.262, 4.895] | 0.029* | 1.399 | [0.971, 2.016] | 0.188 |
| Optimization | 0.950 | [0.668, 1.351] | 0.776 | 0.899 | [0.631, 1.280] | 0.647 | 0.946 | [0.774, 1.156] | 0.772 |
| Other ML | 1.896 | [1.362, 2.640] | 0.002** | 1.563 | [1.114, 2.192] | 0.029* | 0.824 | [0.673, 1.010] | 0.188 |
| Probabilistic | 0.827 | [0.469, 1.459] | 0.567 | 0.781 | [0.441, 1.382] | 0.518 | 0.943 | [0.717, 1.242] | 0.785 |
| Reinforcement Learning | 1.379 | [1.031, 1.844] | 0.071 | 1.010 | [0.751, 1.358] | 0.946 | 0.733 | [0.624, 0.860] | 0.001** |
| Transfer Learning | 1.336 | [0.918, 1.944] | 0.228 | 1.278 | [0.867, 1.884] | 0.378 | 0.956 | [0.755, 1.211] | 0.785 |
| Representation Learning | 1.218 | [0.930, 1.596] | 0.245 | 1.034 | [0.785, 1.361] | 0.854 | 0.848 | [0.716, 1.006] | 0.188 |
| Overall | 0.916 | [0.840, 0.999] | 0.046* | 0.833 | [0.764, 0.909] | 0.000*** | 0.910 | [0.862, 0.960] | 0.001** |

[a] N, D, and H denote *neutral-intended* LLM-assisted, *directional-intended* LLM-assisted, and human-written reviews, respectively. Thus, D−N, D−H, and N−H denote pairwise comparisons between the corresponding groups. Entries report the odds ratio (OR) of acceptance comparing first group to the second group within primary subject group.
[b] $q$-values are from two-sided Wald tests for the model-based log-odds contrasts (reported as odds ratios). We apply Benjamini–Hochberg correction; stars indicate significance after correction (*$q < 0.05$, **$q < 0.01$, ***$q < 0.001$).

*Table B.9.* Joint Wald tests for heterogeneity from the logistic regression model.

| Interaction term | $\chi^2$ | df | $p$ |
|---|---|---|---|
| Review composition condition $\times$ Rating Category | 22.31 | 4 | $< 0.001$*** |
| Review composition condition $\times$ Primary Area | 132.33 | 40 | $< 10^{-11}$*** |

