# OpenReview forum: "Position: Prompting Intent Should Be Audited in LLM-Assisted Peer Review"
_ICML.cc/2026/Position_Paper_Track — ICML 2026 Position Paper Track regular_

### Official Review · Reviewer_8HTe · 2026-03-06

**Significance:** 2
**Argument Clarity:** 4
**Rating:** 3
**Confidence:** 3

**Questions:**

The paper discussed "directional" and "neutral" intents. However, experiments are done with single-directional prompts, i.e. prompts with negative intents. Is there a specific reason not to test positive prompts?

**Alternative Views Section:**

Yes

**Compliance With Llm Reviewing Policy A Conservative:**

Affirmed.

**Discussion Potential:**

3

**Final Justification:**

The authors have answered my questions. My concern is still whether the experiments shown on the paper can demonstrate anything, as the results are not surprising to me at all.  I raised my score to 3 because the authors addresses the rest of my questions.

**Paper Summary:**

This paper argues that reviews that use LLMs to assist their paper reviews should disclose not only if and how much they used LLMs, but also the intent of their prompt. This paper uses synthetic data generated by prompting LLMs to review ICLR 2025 to train a model to infer the review prompt intent, if LLMs were used, and uses this model to analyze the reviews in ICLR 2026. The findings include that directional prompt will likely to incur lower ratings and higher confidence.

**Position:**

Yes

**Position In Title:**

Yes

**Related Work:**

4

**Strengths And Weaknesses:**

This paper reveals an important fact that was majorly neglected by conference policy makers: Intent of the prompt matters. If the prompt contains instructions that does not favor the paper, the resultant review more likely generates negative feedbacks on this paper, therefore leading to a rejection.

However, I feel this paper has some flaws in methodology and viewpoints.

**Methodology**: The authors trains a model to predict the prompt intent from reviews alone, which achieves a pretty good accuracy in section 3.1. This evaluation did not control an important factor: the recommendation score. It is obvious that a negatively directional prompt will lead to a unfavorable review, thus inferring the intent from the review can be largely solved by sentiment analysis.

A more rigorous experiment is to predict the prompt intent conditioned on the review score. For example, suppose we have paper A and B, and A is better than B in fact. Because A is reviewed by a "harshly prompted" AI and B is reviewed by an unbiased AI, both A and B got a score of 5. Can your model tell the prompt intents for paper A and B are different?

Second flaw: The assumption that the intent can be inferred solely from review text ignores too many features.  A biased review might intentionally ignore the advantages of a paper, which cannot be detected solely from the review texts.

**Viewpoint**: The view of the paper is to advocate reviewer to disclose their prompt intent. My alternative view is that very few reviewers would dare to disclose that their prompt is biased.  The authors partially discuss in their alternative view 3, but I think even with non-disclosure terms, the statistics will still be very biased.

**Support:**

2

---

> ### Author Rebuttal · Authors · 2026-03-30
>
> We thank the reviewer for raising important concerns regarding both the methodology and the broader implications of our work. Below, we summarize your comments and provide point-by-point responses.
>
> > W1: The evaluation did not control the recommendation score.
>
> We agree that scores are an informative signal. In this work, we intentionally do not condition on scores, because doing so is *not fully well-posed with available data and may introduce additional bias*.
>
> First, even when a reviewer relies heavily on LLMs, the final score may still be adjusted manually rather than copied directly from the LLM output. A classifier conditioned on score may NOT transfer well from synthetic data to real reviews. Second, in the real ICLR data, the scores provided by the OpenReview API are post-rebuttal and may not fully align with the review text we analyze.
>
> Therefore, we treat score as *an external variable for validation rather than an input for prediction*. This avoids injecting a noisy and potentially misaligned signal into intent detection.
>
> Regarding the *matched-score experiment*, we agree that it could provide additional insight, but it is difficult to validate cleanly in practice for the above reasons. Moreover, the example also illustrates the limitation of score as a proxy: two papers with different quality and prompting conditions can receive the same score. This suggests that score does not reliably capture prompting intent by itself. By contrast, our current evidence shows that *intent can still be meaningfully recovered from review text alone*.
>
> > W2: The assumption that the intent can be inferred solely from review text ignores too many features.
>
> We acknowledge that inferring intent from review text alone may have limitations. More generally, this relates to the broader challenge of grounding and evidence-based reasoning in NLP research. While incorporating richer information, such as the full paper, could improve prediction as the reviewer suggests, it would make the setting computationally intensive and technically complex.
>
> Moreover, this would NOT change the main conclusion. Our framework is in principle compatible with richer inputs, but exploring those extensions is beyond the main scope of this position paper. Our goal is NOT to build the best possible intent predictor, BUT to establish a more basic point: *prompting intent leaves inferable signals in text and therefore can, and should, be audited in peer review*. Our results already provide sufficient evidence that such intent signals are **recoverable even in a text-only setting**.
>
> > W3: The view of the paper is to advocate reviewers to disclose their prompt intent. My alternative view is that very few reviewers would dare to disclose that their prompt is biased.
>
> We respectfully believe this concern is based on *a misunderstanding of our proposal*. We **never advocate reviewers to disclose their prompting intent**. In fact, one motivation for training the detector on synthetic data is to avoid relying on self-reported prompting intent. In practice, such reports raise compliance concerns and may also be incomplete or unreliable.
>
> In Alternative View 3, our discussion of “disclosure” refers to disclosing auditing results back to reviewers, which may create chilling effects, as discussed in Section 6.2. For this reason, we **explicitly recommend NON-disclosure of intent signals to individual reviewers**, minimal-intrusion deployment, and conservative governance-side use.
>
> > Q1: The paper discussed "directional" and "neutral" intents, but the experiments are done with single-directional prompts.
>
> Thank you for your comment. We agree that testing accept-leaning prompts would be valuable. In this paper, we focus on reject-leaning prompts because they represent the most immediate and high-stakes risk in peer review, and are also the most common form of directional prompting. Starting from this setting allows us to isolate a concrete and consequential directional mode in a tractable proof-of-concept setup.
>
> Our goal in this paper is to establish that *prompting intent is detectable and such intent can substantially affect review outcomes*. Once this is demonstrated in a single-directional setting, the framework can be *naturally extended to richer intent taxonomies*, including accept-leaning prompts. However, this extension would likely require more carefully designed synthetic data and additional modeling considerations. While we believe that is an important direction, it is beyond the scope of the current paper and would risk diluting the main message.
>
> In the revision, we will include the above discussion to make this scope choice more explicit and discuss richer intent taxonomies as an important extension of the framework.
>
> We hope our clarifications address the reviewer’s concerns and help better position our contribution as a first step toward understanding and auditing prompting intent in LLM-assisted peer review.

---

> > ### Author Rebuttal · Reviewer_8HTe · 2026-04-01
> >
> > Thanks for clarifying my concerns. I am in doubt whether the experiments can imply something, but the rest of the arguments is valid. I would raise my rating from 2 to 3 thereafter.

---

### Official Review · Reviewer_8dkn · 2026-03-12

**Significance:** 3
**Argument Clarity:** 3
**Rating:** 5
**Confidence:** 3

**Questions:**

See weaknesses above.

**Alternative Views Section:**

Yes

**Compliance With Llm Reviewing Policy A Conservative:**

Affirmed.

**Discussion Potential:**

3

**Paper Summary:**

This paper claims that "what matters is not whether LLMs are used, but how they are instructed." It talks about one of the most challenging issues in current ML community, the peer review system together with the use of LLMs. Using synthetic datasets, a prompt intent detector is designed. With ICLR 2026 real data, some extensive experiments are conducted.

**Position:**

Yes

**Position In Title:**

Yes

**Related Work:**

3

**Strengths And Weaknesses:**

Strengths:
- The topic is timely and super important;
- The motivation is very strong and clear, and i personally like this topic very much, because right now our ML community really needs to think about a transparent and fair reviewing mechanism, especially considering how to 1) make the most use of LLMs to improve the efficiency and also 2) save time and relieve burden for the reviewers;
- A prompt intent detector is designed, with extensive experiments on ICLR 2026.

Weaknesses:
- Although the author claimed that "For tractability, we focus on a single directional mode that steers feedback toward a more critical, rejection-leaning evaluation.", from my view, the more category labels, the more meaningful this task will be. So far, i only see 2 labels (neutral, directional), which is not informative enough. Clearly considering more types of intents can be more helpful, for example, accpet-leaning prompts.
- The detector is based on synthetic data, which might has some gaps from real world.
- In real action, the weights of each label can be output (e.g., how strong it is to be reject-leaning), because it can help to judge the reviewers' action.

**Support:**

4

---

> ### Author Rebuttal · Authors · 2026-03-30
>
> We thank the reviewer for the positive assessment and for highlighting the importance and timeliness of this problem. Below, we summarize your comments in quotes and provide our point-by-point responses.
>
> > W1: Only see 2 intent labels (neutral, directional). Considering more types of intents can be more helpful, for example, accept-leaning prompts.
>
> Thank you for your important comment. We agree that extending to richer intent categories (e.g., accept-leaning prompts) is a helpful direction. In this work, we intentionally focus on a single directional mode (reject-leaning) as a minimal and the most high-impact setting to establish a proof of concept.
>
> Our goal is to demonstrate that **prompting intent is detectable from review text**. Once this signal is shown to be recoverable, the framework can *naturally be extended to multi-class or even continuous intent modeling*. However, doing so would require more carefully designed synthetic data and additional modeling considerations, which are beyond the scope of this paper.
>
> In the revision, we will include the above discussion to make this scope choice more explicit and discuss richer intent taxonomies as an important extension of the framework.
>
> > W2: The detector is based on synthetic data, which might have some gaps from the real world.
>
> We appreciate the opportunity to elaborate this point. We agree that synthetic data may not fully match real-world review distributions, as we mentioned in Section 3.1. However, in this setting, reliable real-world intent labels are extremely difficult to obtain due to both compliance concerns and self-reporting bias. Even if human annotators were asked to infer prompting intent from review text, this would remain a challenging classification problem, since intent is only indirectly reflected in the final text and may be interpreted differently across annotators.
>
> In this paper, we provide **two complementary forms of evidence that a classifier trained on synthetic data can still be informative in the real setting**. First, the leave-one-out experiments across generator LLMs and prompt templates (Table 2 and B.1) demonstrate generalizability, suggesting that the detector is capturing signals that are not tied to one specific model or prompt wording. Second, when applied to real ICLR 2026 reviews, the classifier produces non-trivial and systematic patterns, including meaningful associations with linguistic features (Section 4.1) and decision outcomes (Section 4.3 and 4.4), especially for borderline papers. While this does not provide direct ground-truth validation, it offers indirect evidence that the classification is capturing a meaningful and practically relevant signal rather than synthetic artifacts.
>
> So while we acknowledge the synthetic-to-real gap, we believe synthetic data is currently the most practical way to obtain supervision at scale for this problem. A more realistic path forward is not to eliminate synthetic data, but to improve its diversity, increase modeling robustness, and use the resulting signal in a calibrated and audit-oriented manner.
>
> > W3: In real action, the weights of each label can be output (e.g., how strong it is to be reject-leaning), because it can help to judge the reviewers' action.
>
> We thank the reviewer for this insightful comment. We agree that finer-grained or continuous outputs for each label would provide richer information for auditing. In our current setup, the classifier produces logits, which may serve as a useful proxy for the strength of directional intent. Moreover, the strong $\text{AUC}$ on synthetic data suggests that the model can achieve a reasonable ranking of reviews by directional tendency. This indicates that combining intent classification with confidence calibration is a feasible direction for future work. We will acknowledge in the revision that more fine-grained intent modeling could provide more informative audit signals.
>
>
> We sincerely hope that these clarifications satisfactorily respond to the comments, and welcome any further questions or insights. Thank you once again for your insightful suggestions and valuable feedback!

---

> > ### Author Rebuttal · Reviewer_8dkn · 2026-04-02
> >
> > Thanks for the detailed responses. Personally, I really like the idea of this paper. Considering that i have already given a rather high score, i would like to maintain my positive score 5.

---

### Official Review · Reviewer_M28g · 2026-03-12

**Significance:** 3
**Argument Clarity:** 3
**Rating:** 5
**Confidence:** 4

**Questions:**

- We can very well use ICLR 2026 synthetic reviews to train the intent detector and test ICLR 2026 real reviews. What happens if we do that? Would it change the results?
- What would the human-written distinctive terms look like in Figure 2?

**Alternative Views Section:**

Yes

**Compliance With Llm Reviewing Policy A Conservative:**

Affirmed.

**Discussion Potential:**

3

**Final Justification:**

I am happy with the rebuttal and would keep my positive score. I thank the authors for their rebuttal.

**Paper Summary:**

The paper argues that while LLM-assisted peer-reviewing with usage detection or self-disclosure becomes more common, LLM usage does not fully determine risk. Instead, they posit that the reviews must be audited based on the prompting intent of the reviewers that encodes how the LLM was instructed. Since this intent is latent and unknown, they train an intent detector to find the latent intent from the review text using synthetically labeled LLM-generated reviews. This is trained to classify between neutral and directional intents that differentiate between a stance-neutral generation and a decision-steering generation. They then use this to do an audit on the real ICLR 2026 reviews and find that 20% of flagged reviews are classified as decision-steering. They also conduct word-level, linguistic, structural, and critical semantics analysis to find text-level differences between the directional and neutral reviews. Finally, they also study how acceptance and confidence scores relate to the reviewer's intent scores in different paper qualities and find that directional-intended usage typically leads to a lower rating with a higher confidence trend.

**Position:**

Yes

**Position In Title:**

Yes

**Related Work:**

3

**Strengths And Weaknesses:**

**Strengths**
- The position is very well presented and motivated with a thorough discussion of prior work. Table 1 summarizes the prior policies, and the introduction summarizes the LLM usage and intent literature well.
- The supporting experiments are sound and comprehensive across various models.
- The trained intent detector can be a useful artifact in its own right, but even otherwise, the training methods of the intent detector can be easily extended by conferences to specific data to make it even better.
- The position is timely, given recent incidents, and thus has the potential to spark very interesting discussions.
- They also discuss implications for auditing and actionable suggestions that can be taken by the organizers.

**Weaknesses**
- The paper does not discuss the AAAI auto-reviewing initiative and how it will change the classification strategy. For example, one can generate an intent-specific review for each paper and then compare the submitted reviews with them to come up with a better estimate of the latent intent. This is arguably better than training an intent detector model and may also generate multiple objective scores using LLM-as-a-judge kind of metric.

**Support:**

3

---

> ### Author Rebuttal · Authors · 2026-03-30
>
> We thank the reviewer for the positive evaluation and for recognizing the contributions of our work. Below, we summarize your comments in quotes and provide our point-by-point responses.
>
> > W1: The paper does not discuss the AAAI auto-reviewing initiative and how it will change the classification strategy.
>
> We thank the reviewer for this thoughtful suggestion. We agree that the AAAI auto-reviewing initiative is a relevant context for our discussion, and we will incorporate it into the paper’s review of recent conference policies on LLM-assisted peer review in Table 1 as an easy fix.
>
> We also agree that the AAAI initiative suggests a complementary approach to prompting-intent auditing, especially at the intent-classification stage. In particular, the availability of fully LLM-assisted reviews for each paper could provide an additional reference point for calibrating intent estimates. More broadly, as the reviewer suggests, one could extend this setup by generating multiple intent-specific reviews for each submission and comparing submitted reviews against them.
>
> We view this as a valuable extension within the same overall framework, rather than a replacement for our approach. Our results already show that prompting intent is recoverable from review text, which directly supports our central position. While such generation-based approaches may offer finer paper-specific calibration, they also introduce additional computational and procedural complexity, since they require generating multiple LLM-assisted reviews before classification. This burden would grow further if the space of intent categories were expanded beyond the current setting. Our paper intentionally adopts a minimal and tractable setup to establish this core claim. In deployment, one could certainly consider richer alternatives, including generation-based comparisons or selectively invoking such procedures when classification is difficult or uncertainty is high. However, **these choices do NOT change our central position, but rather expand the design space within the auditing framework we advocate**.
>
> > Q1: We can very well use ICLR 2026 synthetic reviews to train the intent detector and test ICLR 2026 real reviews. What happens if we do that? Would it change the results?
>
> Thank you for your valuable questions. We intentionally avoid generating synthetic reviews from ICLR 2026 submissions to **prevent potential data leakage**. Using the same set of papers for both synthetic training and real-world evaluation could introduce unintended correlations, making the evaluation less reliable.
>
> By training on ICLR 2025 and evaluating on ICLR 2026, we ensure a cleaner separation between training and testing distributions, which provides a more rigorous assessment of generalization. We will include the above discussion on this design choice in the revision.
>
> > Q2: What would the human-written distinctive terms look like in Figure 2?
>
> We thank the reviewer for this helpful question. In response, we **applied exactly the same distinctiveness analysis as in Figure 2 to compare human-written and AI-generated reviews**, and report the top 10 terms in table form below.
>
> The resulting pattern is clear: AI-generated reviews are characterized more by broad evaluative and technical language, suggesting a more standardized and high-level academic framing. In contrast, human-written reviews contain more manuscript-grounded and reviewer-voice expressions, indicating closer engagement with the specific paper, its structure, and more personalized judgment.
>
> | AI-generated | ||||
> |-----------|-----------|----------|-----------|-------------|
> |reasoning | framework | analysis | empirical | theoretical |
> | vs        | practical | real     | robustness| gains       |
>
> | Human-written | ||||
> |-----------|-----------|----------|-----------|-------------|
> | proposed  | authors   | work     | section   | arxiv       |
> | line      | interesting| used    | think     | good        |
>
>
> We sincerely hope that these clarifications and additional results satisfactorily respond to the comments, and welcome any further questions or insights. Thank you once again for your thoughtful feedback and helpful questions!

---

> > ### Author Rebuttal · Reviewer_M28g · 2026-04-03
> >
> > I am happy with the rebuttal and would keep my positive score. I would encourage the authors to add these discussions and clarifications to the camera-ready version. Good luck!

---

### Official Review · Reviewer_m1WR · 2026-03-14

**Significance:** 4
**Argument Clarity:** 4
**Rating:** 5
**Confidence:** 4

**Questions:**

- Why did you use JINA-EMBEDDING-V3 as your intent classifier?
- How did you define the categories in Table A.3? Are there any possibilities of causing some biases in the interpretation of the results due to this categorization?

**Alternative Views Section:**

Yes

**Compliance With Llm Reviewing Policy A Conservative:**

Affirmed.

**Discussion Potential:**

4

**Final Justification:**

I don't have any strong concerns about this paper. So, I keep my current high score.

**Paper Summary:**

In this paper, the authors discuss the current issue of the LLM-assisted peer review, focusing on the biases caused by its prompt intention. Specifically, such reviews are classified into stance-neutral LLM-generated reviews and stance-primed generation aimed at a presumed outcome. The authors reveal the existence of this bias through analyses using a newly constructed intent classifier. Based on the analysis, the authors provide some recommendations to mitigate the issues.

**Position:**

Yes

**Position In Title:**

Yes

**Related Work:**

3

**Strengths And Weaknesses:**

Strengths:
- As well as listing the authors' claims, this paper provides specific experiments to confirm the validity of the claims.
  - The authors show that prompting intent is recoverable from LLM-generated review text using the authors' created intent classifier.
  - The authors reveal that 20.51% of ICLR2026 reviews are classified as directional using the intent classifier.
- The word-level analysis shows the obvious tendency of the word use between neutral and intent reviews.
- The analysis characterizes stylistic differences of reviews that the intended reviews allocate a substantially larger share of content to critical sections. Supporting this intuitive tendency, relying on a quantitative analysis is a large contribution.
- Furthermore, the analysis based on the predefined keywords and phrases shows that directional-intended reviews have higher frequencies in most criticism categories.
- The suggested recommendations for solving the current issues are concrete and practical.

Weaknesses:
- The suggested recommendations are not fundamental solutions. However, it's not a strong weakness as a position paper.

**Support:**

4

---

> ### Author Rebuttal · Authors · 2026-03-30
>
> We thank the reviewer for the positive evaluation and for recognizing the contributions of our work. Below, we summarize your comments in quotes and provide our point-by-point responses.
>
> > Q1: Why use JINA-EMBEDDING-V3 as your intent classifier?
>
> Thank you for your important comment. We use JINA-Embedding-V3 because it is a relatively recent embedding model, so its training corpus is likely more up to date and better able to capture newer terminology that appears in research papers. It also performs strongly across multiple benchmarks.
>
> Since the encoder is used only as a generic feature extractor, we expect the main findings *not to depend on a particular embedding model*, as long as the encoder provides sufficiently strong semantic representations. To verify this explicitly, we **re-ran the synthetic experiment using four widely adopted embedding models**. As different encoders produce probability scores on different scales, we calibrate the classification threshold for each encoder independently on a held-out validation set by maximising macro F1, ensuring a fair comparison. Across all embedding choices, we observe highly consistent predictive performance ($\text{ROC-AUC} \geq 0.90$ for all models), and we further confirm strong agreement in the resulting classifications across models, with $\text{Fleiss' }\kappa=0.67, \text{Kendall's }W=0.82$, and mean pairwise $\text{Spearman }\rho=0.76$. This indicates that *our findings are not driven by a particular embedding space*, but instead reflect **a robust and largely model-agnostic signal in the review text**.
>
> | Embedding Model | Threshold | Accuracy  | ROC-AUC | PR-AUC | TPR  | FPR   | F1 |
> |---|---|---|---|---|---|---|---|
> | jinaai/jina-embeddings-v3 | 0.54 | 0.9407 | 0.9834 | 0.9835 | 0.9185 | 0.0370 | 0.9407 |
> | intfloat/e5-large-v2 | 0.50 | 0.9037 | 0.9546 | 0.9620 | 0.9185 | 0.1111 | 0.9037 |
> | Snowflake/snowflake-arctic-embed-l | 0.50 | 0.8148 |  0.9004 | 0.9202 | 0.8000 | 0.1704 | 0.8148 |
> | BAAI/bge-m3                            |   0.45 | 0.8704 |  0.9457 | 0.9498 | 0.8963 | 0.1556 | 0.8703 |
>
>
> > Q2: How to define the categories in Table A.3 and potential bias?
>
> We appreciate the opportunity to elaborate this point. The categories in Table A.3 are defined post hoc to provide an interpretable and transparent breakdown of observed criticism patterns, rather than to drive the main analysis or impose a canonical taxonomy.
>
> We acknowledge that lexicon-based categorization may introduce bias in how specific linguistic patterns are grouped. However, this analysis is descriptive and supplementary, and *our main conclusions do NOT depend on any single category*. The key findings are supported by consistent patterns across multiple dimensions and independent outcome-based analyses (e.g., ratings, confidence, and acceptance).
>
> > W1: The suggested recommendations are not fundamental solutions. However, it's not a strong weakness as a position paper.
>
> With many conferences, including ICML, allowing the use of LLMs to assist peer review, it is becoming increasingly important to understand **how** LLMs are used in the interest of academic integrity, transparency, and accountability. This motivates our central position.
>
> We agree that the recommendations outlined in the paper are not intended as a complete or definitive solution. At the current stage, it is difficult to specify a sufficient solution that accommodates the interests of multiple stakeholders. Bridging these gaps and reducing bias will require substantially more complex investigation, including a deeper understanding of both linguistic and social structures.
>
> Our goal in this position paper is therefore modest but necessary: to formalize and empirically validate that prompting intent is a measurable latent factor, and to introduce an intent-centric auditing framework as an initial step toward this broader goal. We acknowledge in the limitations that inferred intent signals are only proxies and should be treated as calibrated risk indicators rather than ground-truth judgments, and that intent auditing itself may introduce risks, including chilling effects on reviewer expression.
>
> We also note that the community is only beginning to explore controlled experiments around LLM-assisted reviewing. In this setting, solutions that balance transparency, accountability, reviewer autonomy, and practical feasibility will likely need to emerge iteratively rather than in a single step. We hope our paper helps establish part of the empirical and conceptual foundation for that process.
>
>
> We sincerely hope that these clarifications and additional results satisfactorily respond to the comments, and welcome any further questions or insights. Thank you once again for your thoughtful feedback and helpful questions!

---

> > ### Author Rebuttal · Reviewer_m1WR · 2026-04-04
> >
> > The authors answered my questions, and that resolved my concerns. The weakness I pointed out is still there, but it's not a large issue for a position paper. So, I keep this high score.

---

### Decision · Program_Chairs · 2026-04-30

**Decision:**

Accept (regular)

**Comment:**

This is a timely and well-argued position paper on an issue the community is already confronting. The main claim is clear, and the paper goes beyond opinion by providing empirical evidence that prompting intent leaves detectable signals in review text and may be associated with review outcomes.

The reviews were overall positive. The main concern was about the limits of the current empirical setup, including the use of synthetic data, the focus on one directional mode, and whether text alone is sufficient to infer intent. These are valid limitations, but after rebuttal they were largely clarified, and no reviewer raised a remaining objection strong enough to outweigh the paper’s contribution.

Overall, I find it states a clear position, contrasts it with non-trivial alternatives, and supports it with enough evidence to make the discussion concrete and useful.